# Development of Bedaquiline-Loaded SNEDDS Using Quality by Design (QbD) Approach to Improve Biopharmaceutical Attributes for the Management of Multidrug-Resistant Tuberculosis (MDR-TB)

**DOI:** 10.3390/antibiotics12101510

**Published:** 2023-10-03

**Authors:** Rao Nargis Jahan, Zafar Khan, Md. Sayeed Akhtar, Mohd Danish Ansari, Pavitra Solanki, Farhan J. Ahmad, Mohd Aqil, Yasmin Sultana

**Affiliations:** 1Department of Pharmaceutics, School of Pharmaceutical Education and Research, Jamia Hamdard, New Delhi 110062, India; rnjahan@jamiahamdard.ac.in (R.N.J.); danishjamia786@gmail.com (M.D.A.); fjahmad@jamiahamdard.ac.in (F.J.A.);; 2Department of Clinical Pharmacy, College of Pharmacy, King Khalid University, Al-Fara, Abha 62223, Saudi Arabia; 3Department of Pharmaceutics, Delhi Pharmaceutical Sciences and Research University, New Delhi 110017, India; drpavitra@dpsru.edu.in

**Keywords:** bedaquiline, oral bioavailability, solubility, quality by design (QbD), pseudo-ternary, self-nanoemulsifying drug delivery systems (SNEDDS), multi-drug resistance tuberculosis (MDR-TB)

## Abstract

**Background:** The ever-growing emergence of antibiotic resistance associated with tuberculosis (TB) has become a global challenge. In 2012, the USFDA gave expedited approval to bedaquiline (BDQ) as a new treatment for drug-resistant TB in adults when no other viable options are available. BDQ is a diarylquinoline derivative and exhibits targeted action on mycobacterium tuberculosis, but due to poor solubility, the desired therapeutic action is not achieved. **Objective:** To develop a QbD-based self-nanoemulsifying drug delivery system of bedaquiline using various oils, surfactants, and co-surfactants. **Methods:** The quality target product profile (QTPP) and critical quality attributes (CQAs) were identified with a patient-centric approach, which facilitated the selection of critical material attributes (CMAs) during pre-formulation studies and initial risk assessment. Caprylic acid as a lipid, propylene glycol as a surfactant, and Transcutol-P as a co-surfactant were selected as CMAs for the formulation of bedaquiline fumarate SNEDDS. Pseudo-ternary phase diagrams were constructed to determine the optimal ratio of oil and S_mix_. To optimize the formulation, a Box–Benkhen design (BBD) was used. The optimized formulation (BDQ-F-SNEDSS) was further evaluated for parameters such as droplet size, polydispersity index (PDI), percentage transmittance, dilution studies, stability studies, and cell toxicity through the A549 cell. **Results:** Optimized BDQ-F-SNEDDS showed well-formed droplets of 98.88 ± 2.1 nm with a zeta potential of 21.16 mV. In vitro studies showed enhanced drug release with a high degree of stability at 25 ± 2 °C, 60 ± 5% and 40 ± 2 °C, 75 ± 5%. Furthermore, BDQ-F-SNEDDS showed promising cell viability in A549 cells, indicating BDQ-F-SNEDDS as a safer formulation for oral delivery. **Conclusion:** Finally, it was concluded that the utilization of a QbD approach in the development of BDQ-F-loaded SNEDDS offers a promising strategy to improve the biopharmaceutical properties of the drug, resulting in potential cost and time savings.

## 1. Introduction

The ever-growing emergence of antibiotic resistance associated with tuberculosis (TB) has become a global challenge [1]. Furthermore, the combination of TB and COVID-19 can cause severe respiratory complications, leading to systemic failure [2]. According to reports from the World Health Organization (WHO), TB was estimated to be the leading cause of death from a single infectious agent and the thirteenth leading cause of death worldwide [3]. However, in 2020 and 2021, it was projected that TB would rank second as a cause of death from a single infectious agent following the emergence of COVID-19 [4]. In 2021, it was estimated that approximately 10.6 million individuals worldwide contracted tuberculosis (TB); out of this total, it is estimated that 191,000 deaths occurred specifically due to multidrug-resistant or rifampicin-resistant tuberculosis (MDR/RR-TB) [5].

For the management of MDR-TB, WHO suggests two-course durations, which are abbreviated as short (9–11 months) and long (18–20 months) treatment regimens [2,6]. However, the concomitant use of multiple anti-TB medications and lengthy treatment regimens is often met with poor patient adherence, resulting in the emergence of Mycobacterium strains, which are resistant to first-line drugs such as isoniazid and rifampicin, leading to one of the most serious kinds of tuberculosis, i.e., MDR-TB. This has spurred the necessity of research and development of new drugs to fight against TB [7,8]. Albeit, various regulatory bodies are working to overcome the burden of this disease. The USFDA gave expedited approval to bedaquiline (TMC207) in 2012 as a new treatment for drug-resistant TB in adults when there are no other viable options available [9].

Bedaquiline (BDQ), a diarylquinoline derivative, has a unique mechanism of action. The drug specifically targets and deactivates the F1/F0-ATP synthase of MDR strains of *Mycobacterium tuberculosis (M. tb)* by binding to its subunit C of ATP synthase enzyme, which is responsible for energy production in *M. tb* without interfering with the F1/F0-ATP synthase in mammalian cells [10,11]. In preclinical and clinical studies, bedaquiline has shown promising results in treating MDR-TB. These studies involved the use of bedaquiline as a single drug or in combination with other anti-TB drugs (e.g., moxifloxacin and pyrazinamide). Additionally, some studies utilized a murine model of TB, and the findings support the efficacy of bedaquiline in treating MDR-TB [12]. In these studies, bedaquiline was administered at a dosage of 400 mg once daily for a period of 2 weeks, followed by 200 mg three times weekly for either 6 or 22 weeks, and showed improved treatment outcomes when compared with existing treatment regimens. Also, the bedaquiline regimen led to faster culture conversion (when the bacteria are no longer detectable in the patient’s sputum); this suggests that treatment duration with bedaquiline regimens may be shorter compared to other standard regimens [13].

Despite this unique anti-TB action of this drug, it is also associated with certain physiochemical limitations. According to the Biopharmaceutical Classification System (BCS), bedaquiline belongs to the Class II category, as it has exceptionally low solubility in water (0.000193 mg/mL) and high lipophilicity (log P 7.1) [14], thus making it an unsuitable molecule for conventional dosage forms. Furthermore, bedaquiline carries with it several dose-dependent potential adverse effects, such as prolonged QT interval, increased hepatic aminotransferase levels, nausea, vomiting, etc., which limit its use [15]. There are several formulation strategies, such as reduction in size, complexation, polymorphism, and solid dispersions, which overcome the hurdles of dose-dependent adverse effects, solubility, and release rate but do not resolve the bioavailability problems stemming from gastric degradation.

Of late, researchers have found self-nano-emulsifying drug delivery systems (SNEDDSs) to be the most advantageous lipid-based drug delivery systems for circumventing the existing challenges. They have also gained tremendous popularity due to their advantageous features, like small globule size, simplified production process, increased biocompatibility, and enhanced stability [16]. These systems are typically composed of lipids (natural and synthetic oils), surfactants, co-surfactants, and/or co-solvents that spontaneously emulsify. By pre-dissolving the drug in a mixture of lipids and emulsifying excipients, the disintegration and dissolution steps can be bypassed, which usually hinders the oral absorption of water-insoluble drugs [17].

The systematic use of design of experiments (DoE) to optimize isotropic systems has become common practice in both industry and academia [18]. The recent approach of “formulation by design” (FbD), based on DoE and quality by design (QbD), provides a rational understanding of the interaction between variables and helps to select the best formulation with minimal time, effort, and cost compared to the traditional one-factor-at-a-time (OFAT) approach [18]. The FbD methodology involves defining the quality target product profile (QTPP), identification of critical quality attributes (CQAs), critical material attributes (CMAs), and critical process parameters (CPPs) through screening and risk assessment, optimization data analysis using DoE, modeling and optimum search through response surface methodology (RSM) to create the design space, and developing a control strategy for continuous improvement [19]. The previous experimental studies carried out in various laboratories on diverse nanostructured systems, like liquid and S-SNEDDS of carvedilol [20] and ezetimibe [21], solid lipid nanoparticles of quercetin [22], isotretinoin [23], and liposomes of nimesulide [24], have vouched for the utility of QbD in developing the optimized nanocolloidal formulations.

The novelty of the current research work is related to the preparation of SNEDDS loaded with bedaquiline fumarate through the systematic optimization of the concentration of oil, surfactant, and co-surfactant. Furthermore, the study aims to investigate the biopharmaceutical performance of the developed SNEDDS, including in vitro release and stability studies, to confirm their retainability and quality under various physiological conditions. Finally, to assess the cytotoxicity of the optimized formulation, an MTT assay was conducted on A549 cells.

## 2. Results and Discussion

### 2.1. Solubility Study

The maximum solubility of BDQ-F in oils was found in caprylic acid (6.27 ± 10.90 mg/mL) and among the surfactants and co-surfactants, propylene glycol and Transcutol-P exhibited the highest solubility (6 ± 2.10 mg/mL), as shown in Figure 1 and Figure 2. Caprylic acid is a medium-chain saturated fatty acid, and the free fatty acid significantly increases the solubility of the drug by showcasing it as a lipophilic solubilizer [25]. Propylene glycol is known to reduce the interfacial tension of oil in water [26]. Transcutol-P has an HLB value of 4.2 and the ability to permeate GI mucosa by disrupting the lipid bilayer and enhancing oral bioavailability.

### 2.2. Pseudoternary Phase Diagrams

Pseudoternary phase diagrams were created to determine the optimal surfactant: co-surfactant (S_mix_) and oil: S_mix_ ratios for SNEDDS development, as shown in Appendix A. The pink dots represent the nanoemulsion region. It was observed that when a 1:0 S_mix_ ratio was used, the nanoemulsion region obtained was almost negligible because propylene glycol was used alone, and sufficient emulsification of oil was not achieved. When the co-surfactant was incorporated in a S_mix_ ratio of 1:1, the nanoemulsion region increased slightly. On increasing the quantity of co-surfactant in the S_mix_ ratio 1:2 further, no increase in the nanoemulsion region was observed. On the other hand, when the concentration of surfactant quantity was increased from 1:1 to 2:1, an increase in the nanoemulsion region was observed, and 13.71% (*v*/*v*) oil was found to be emulsified with 41 % *v*/*v* of S_mix_. On further increasing the quantity of surfactant in the S_mix_ 3:1 ratio, there was an appreciable increase in the nanoemulsion region, and the maximum amount of oil that could be emulsified was found to be 12.90 % *v*/*v* using 51.60 % *v*/*v* of S_mix_. For the S_mix_ 4:1 and 5:1 ratios, there was no appreciable increase in the nanoemulsion region. Based on the pseudoternary diagram concentrations of oil, S_mix_ was taken between the ranges of 10–30% and 40–60% [27]. 

### 2.3. Optimization Using BBD 

The final optimization was conducted using Design Expert^®^ software version 10.0.4 by Stat-Ease in Minneapolis, USA) using caprylic acid (A) (as oil), S_mix_ (B) (consisting of propylene glycol, as a surfactant), Transcutol-P (as a co-surfactant), and the sonication time (C) as independent variables. The size of droplets (measured in nanometers), PDI, and percentage of transmittance were taken as dependent variables. Responses to fourteen runs, as shown in Appendix A, were used to determine the significant model. The polynomial equations obtained for droplet size, PDI, and transmittance are as follows: Droplet size (nm) = + 89.15 + 20.19A − 6.01B + 4.70C + 0.1125AB + 0.5500AC + 1.00BC + 15.36A^2^ + 0.0563B^2^ + 6.14C^2^(1)
PdI = + 0.2050 + 0.0757A − 0.0570C − 0.0347AB + 0.0617AC − 0.0328BC + 0.0129A^2^ − 0.0101B^2^ + 0.0329C^2^
(2)
Transmittance = +96.85 − 2.19A + 0.7375B + 0.3000C − 0.0750AB − 0.1000AC − 0.0500BC − 0.8625A^2^ − 0.0625B^2^ − 0(3)

#### 2.3.1. Effect of Independent Variables on Droplet Size

Droplet size is an important parameter for the drug release. Smaller droplet size corresponds to a larger surface area, which in turn leads to more drug release and, eventually, higher bioavailability. Droplet size increased from 83.1 nm to 135.7 nm with various variables. The amount of oil has a significant positive effect on droplet size, as seen in the 3D response curves (Figure 3). On the contrary, S_mix_ has a negative effect on droplet size. As the concentration of S_mix_ increases from 40% to 60%, droplet size gradually decreases. A larger surfactant concentration decreases the surface tension between oil and aqueous phase, leading to smaller droplet size [28]. An increase in sonication time initially increases the droplet size, but further sonication causes aggregation of droplets and an increase in size. 

#### 2.3.2. Effect of Independent Variables on PdI

The PDI is an indicator of the homogeneity or uniformity of a formulation. In this study, the effects of different variables on the PDI of BDQ-F-SNEDDS were examined using a polynomial equation. As the concentration of lipids and sonication time increased, the PDI also increased significantly. On the other hand, S_mix_ had an inverse effect on the PDI of BDQ-F-SNEDDS. The PDI values ranged from 0.11 to 0.44, indicating variations in droplet size distribution within the formulation.

#### 2.3.3. Effect of Independent Variables on Transmittance

The percentage transmittance in the runs ranged from 93.1% to 98.9%, which is indicative of the transparency of the formulation. Oil percentage has a significant negative impact, whereas S_mix_ and Sonication time has a slight positive effect on transmittance. Higher transmittance indicates the transparency and stability of the formulation, while a lower value shows turbidity and non-homogenous formulation. 

### 2.4. Validation and Point Prediction

On the basis of observed values of variables and the constraints applied, the software generated a predicted response for the optimal formulation, and this response was then compared with the experimental/observed value. The software generated value was comprised of 20% oil, 40% S_mix_, and 30 s sonication time. The predicted responses for droplet size, PDI, and Transmittance were 101 nm, 0.29, and 98.17% respectively. The observed response was 98.88 nm droplet size, 0.32 PDI, and 98.12% transmittance. The predicted and observed values were in good correlation, as seen in Appendix A, establishing the accuracy of BDQ-F-SNEDDS.

### 2.5. Characterization

#### 2.5.1. Droplet Size, PdI, and Viscosity

The droplet size was found to be 98.88 ± 2.10 nm, indicating the average diameter of the droplets in the formulation, which is represented in Figure 4A. The PDI value of 0.3 ± 0.09 suggests a relatively narrow size distribution of the droplets, as represented in Figure 4A. The viscosity of BDQ-F-loaded SNEDDS was found to be 45.30 ± 0.062 cP, and it was observed that an SNEDDS with lower viscosity tends to form an o/w type of nanoemulsion system. 

#### 2.5.2. Thermodynamic Stability Studies

The optimized BDQ-F-SNEDDS formulation displayed excellent stability characteristics, with no indications of instability such as creaming, cracking, or phase separation observed. This investigation demonstrated that the resulting formulation remained robust even when subjected to heating, cooling, and freeze–thaw temperature fluctuations even after centrifugation. Moreover, it exhibited no evidence of precipitation or phase separation when exposed to diverse stress conditions. Therefore, the optimized formulation can be deemed thermodynamically sound and stable.

#### 2.5.3. Zeta Potential

BDQ-F-SNEDDS depicted a potential of 21.16 ± 3.4 mV, as shown in Figure 4B. The positive zeta potential indicates that the surface charge of droplets is positive [29]. A formulation is considered stable if the zeta potential value is between +30 mV and −30 mV and no separation or coagulation is seen [30].

#### 2.5.4. Transmission Electron Microscopy

TEM image displays visual evidence of spherical-shaped droplets with an average range of size of 80 to 100 nm, as shown in Figure 5. 

#### 2.5.5. Entrapment Efficacy

BDQ-F-SNEDDS exhibited a high drug loading efficacy of 81.2 ± 1.3%. This can be attributed to the lipophilic nature of the drug, which enhances its solubility in the oil phase, resulting in a higher entrapment efficacy within the formulation.

#### 2.5.6. Self-Emulsification Time

BDQ-F-SNEDDS were completely dissolved within 15 ± 3 s upon gentle agitation, which indicates the ability for ease of emulsification. The low emulsification time may be due to reduced interfacial tension by propylene glycol, which causes diffusion of the aqueous phase into the oil phase [31]. The sonication time also played a much greater effect on the self-emulsification time, as a small change in the sonication time can have a significant impact on the time it takes for the SNEDDS to form a nanoemulsion. It could be due to the reason that sonication breaks down the oil droplets into smaller droplets, which makes it easier for them to be dispersed in the aqueous phase. The longer the sonication time, the smaller the oil droplets will be, and the faster the SNEDDS will form a nanoemulsion.

#### 2.5.7. Dilution Study 

Optimized BDQ-F-SNEDDS were found to be robust at various dilutions. Results shown in Table 1 depict that BDQ-F-SNEDDS were stable at different dilutions. When the samples were diluted, a slight reduction in droplet size and transmittance was observed. However, the PDI increased as a result of the dilutions.

#### 2.5.8. In Vitro Drug Release

Figure 6 illustrates the percentage cumulative drug release (%CDR) of the optimized BDQ-F-SNEDDS and BDQ-F suspension. Within the first 2 h, BDQ-F-SNEDDS released 27% of the drug, whereas after 12 h, 93% of the drug was released. In comparison, the suspension only released 55% of the drug. The improved release of the drug can be attributed to the rapid formation of a nanoemulsion with nanosized droplets. This suggests that the drug was effectively solubilized in the selected lipid and surfactant components of the formulation [32].

#### 2.5.9. Stability Study

Optimized BDQ-F-SNEDDS were evaluated on the basis of physical appearance, phase separation, caking, droplet size, PDI, and entrapment efficiency, as shown in Table 2. When BDQ-F-SNEDDS were kept at a temperature of 40 ± 2 °C and humidity of 75 ± 5% for 6 months, no phase separation was observed, and the physical appearance was also clear. Droplet size and PDI increased from 98.88 nm and 103.04 nm and 0.34 ± 0.392 to 0.45 ± 0.43, respectively. Droplets tend to coalesce when kept for a longer period, resulting in a larger diameter. Entrapment efficiency reduced from 98.31% to 76.20%. Leaking of the drug over the time period leads to a decreased entrapment efficiency of the drug. When kept at a temperature of 25 ± 2 °C and humidity of 60 ± 5%, optimized BDQ-F-SNEDDS droplet size and PdI increased from 98.88 nm to 106.21 nm and 0.34 to 0.48. Optimized BDQ-F-SNEDDSs were stable at both temperatures, indicating a robust formulation.

#### 2.5.10. Cell Cytotoxicity 

The MTT assay was employed to evaluate the cytotoxicity of optimized BDQ-F-SNEDDS, BDQ-F, and a control (placebo) in A549 cells Figure 7. The IC_50_ (the concentration of drug required to inhibit cell growth by 50%) of free BDQ-F was 48.05 ± 1.47 μm. The IC_50_ of optimized BDQ-F-SNEDDS was significantly lower, at 17.49 ± 2.38 μm (*p* < 0.0001). This notable difference could be attributed to the improved solubility of BDQ-F in SNEDDS and the rapid internalization of SNEDDS by cells [33]. 

## 3. Materials

Bedaquiline fumarate (BDQ-F) (99.5% pure) was obtained as a complimentary sample from Omgene Life Sciences Pvt Ltd. (Vadodara, India). The high-performance liquid chromatography (HPLC) grade solvents, such as acetonitrile, methanol, and ethanol, were procured from Merck Ltd. in Mumbai, India. Transcutol-P and propylene glycol were acquired from Sigma-Aldrich (Bangalore, India). All other chemicals used in the study were of analytical grade, and deionized water was used for all experiments. 

## 4. Methods

### 4.1. HPLC Method

The HPLC technique for analyzing BDQ was established following a documented procedure. In brief, an HPLC system (Shimadzu VP, Kyoto, Japan) was employed, featuring a C-18 column (250 × 4.6 mm, with 5 μm particle size), a binary pump, a UV-VIS detector, and Class VP software for data analysis. The mobile phase consisted of a mixture of water (0.01% TFA) and acetonitrile (0.01% TFA) in a ratio of 10:90 *v*/*v*, utilizing analytical grade solvents and filtered through a 0.45 μm membrane filter. The flow rate of the mobile phase was set at 1 mL/min, and the detector was configured to a wavelength of 265 nm [34]. This HPLC methodology served for assessing both the entrapment efficiency and in vitro drug release.

### 4.2. Solubility Studies

#### 4.2.1. Selection of Oil

To determine the solubility of BDQ-F in different oils, the shake flask technique was employed. Excess BDQ-F was added to 2 mL of various oils, including olive oil, fish oil, castor oil, sesame oil, soybean oil, peanut oil, almond oil, and caprylic acid. The mixture was vigorously vortexed for 10 min to ensure thorough mixing. Subsequently, the samples were incubated for 72 h in a rotary shaker bath at a controlled temperature of 37 ± 5 °C to attain equilibrium. After the equilibrium period, the samples were centrifuged at 100 rpm for 5 min. The resulting supernatant was separated, appropriately diluted, and subjected to analysis to determine the concentrations of BDQ-F [35]. 

#### 4.2.2. Selection of Surfactant and Co-Surfactant

Surfactant and co-surfactant selections were based on the emulsification ability. Various surfactants, e.g., tween 80, tween 40, span 80, cremophore, capryol 90, capmul GMO, captex 355, caprol ET, and propylene glycol, were screened for their emulsification capability. 

### 4.3. Pseudo-Ternary Phase Diagram

A pseudoternary phase diagram was built for the SNEDDS development to compute the surfactant: co-surfactant (S_mix_) and oil: S_mix_ ratios. S_mix_ was taken in the following ratios: 1:0, 1:1, 1:2, 2:1, 3:1, 4:1, 5:1. The oil and S_mix_ mixtures were then titrated against deionized water. The amount of deionized water was changed at 5% intervals between 5 and 95%. After adding deionized water to the mixture, it was vortexed for 2 to 5 min and visually examined for clarity or turbidity, with the results noted on the phase diagram. The S_mix_, oil, and deionized water phases of the formulation are represented by the three vertices of the pseudoternary phase diagram. In the pseudoternary phase diagram, the percentage composition of each nanoemulsion was designated as a point, and the area surrounded by these points was measured [36]. 

### 4.4. Quality by Design (QbD) Approach Incorporation

Designing for quality gives value and quality to the final pharmaceutical product. Throughout the formulation development process, it is vital to identify and regulate the critical parameters related to the process and formulation development. The QbD technique entails establishing the quality target product profile (QTPP); identifying critical quality attributes (CQAs), critical material attributes (CMAs), and critical process parameters (CPPs) through screening and risk assessment; optimizing data using DoE; and evaluating the results. Appendix A provide various parameters of QTPP and CQAs for SNEDDs formulation. 

### 4.5. Risk Assessment Using Ishikawa

A risk assessment approach was developed to detect and estimate the likelihood of drug excipient interactions with various unit functions if there are any risks or failures. The Ishikawa fish-bone diagram was constructed employing Minitab 16 software, as shown in Appendix A. 

### 4.6. Formulation and Optimization of SNEDDS Using Box Behnken Design

Box–Behnken experimental design (BBD) (Design Expert^®^ software version 10.0.4 by Stat-Ease in Minneapolis, Kansas, KS, USA) was used to evaluate the critical experimental conditions for the maximum and minimum response. BBD was used to optimize solid-nanoemulsion drug delivery systems (SNEDDS) by studying the effects of variables on size, entrapment efficiency, and drug release. This design helps to eliminate non-significant factors that do not have a significant effect on the response variable, and it can provide an optimized formulation with fewer trials, consequently leading to cost and time savings. Furthermore, this design simplifies the experiments by reducing the number of variables that need to be considered [37]. SNEDDSs were created after selecting the surfactant and co-surfactant ratio (S_mix_). Using a vortex, an exactly weighed quantity of drug was entirely dissolved in oil. S_mix_ was slowly added drop by drop to the oil drug combination while constantly swirling to generate uniform formulation. After that, the mixture was sonicated to achieve the desired droplet size. BBD is a more suitable approach for evaluating the effects of formulation variables and the effect on their corresponding variables [38]. The goal of this optimization was to investigate how different independent variables influenced dependent variables. These factors included caprylic acid (A) used as an oil, ranging from 10 to 30 mL; S_mix_ (B), consisting of propylene glycol as a surfactant, and Transcutol-P as a co-surfactant, ranging from 40 to 60 mL. The sonication time varied from 30 to 60 s. The dependent variables were the size of droplets (measured in nanometers), the polydispersity index (PDI), and the percentage of transmittance. The levels of independent variables, as well as the constraints of dependent variables, are presented in Table 3. Fourteen runs were generated with two center points with a quadratic polynomial equation.

### 4.7. Characterization of SNEDDS

#### 4.7.1. Droplet Size, PdI, Zeta Potential, and Viscosity

The average diameter of droplet, PdI, and zeta potential was estimated using zeta sizer (Malvern Zetasizer Ver. 7.12, Malvern Instruments, Malvern, UK). Samples were diluted up to 100 times before analysis at 25 °C temperature. 

Viscosity was determined with the help of Brookfield viscometer, Rheoplus Anton paar MCR10. The test was performed in a clean environment at room temperature. Analyses were conducted in triplicate.

#### 4.7.2. Thermodynamic Stability Studies

Thermodynamic stability studies were conducted to evaluate the physical resilience of BDQ-F-SNEDDS. The assessment encompassed factors such as creaming, cracking, phase separation, and precipitation, which were examined through a series of three cycles involving heating, cooling, centrifugation, and freeze–thaw processes. Specifically, the BDQ-F-SNEDDS underwent six cycles spanning temperatures from 4 to 40 °C. Following this, the mixture underwent centrifugation at 2000 rpm for 15 min. Subsequently, it was subjected to three freeze–thaw cycles alternating between temperatures of −21 °C and +25 °C. The evaluation of physical stability was conducted through visual observations [39]. 

#### 4.7.3. Transmission Electron Microscopy (TEM)

Morphological analysis of the prepared formulation was optimized by TEM. To ensure accurate results and avoid any size discrepancies, the sample preparation was performed in a controlled environment free from particles. The procedure involved placing a copper grid on a paraffin sheet using forceps. Subsequently, a diluted sample of BDQ-F-loaded SNEDDS was carefully dropped onto the copper grid using a micropipette. The grid, along with the sample, was then immersed in a 2% phosphotungstic acid solution to enhance contrast. Excess fluid was removed by placing the copper grid on Whatman filter paper. After complete drying of the grid, it was subjected to light beam, and the morphology of droplets was observed under TEM (CM 200, Philips Briacliff Manor, New York, NY, USA).

#### 4.7.4. Entrapment Efficacy

The drug loading efficiency of bedaquiline fumarate of the optimized formulation was determined by centrifugation method. BDQ-F-SNEDDSs were centrifuged at 4000 rpm for 15 min. The supernatant was diluted, filtered, and further analyzed using an established high-performance liquid chromatography (HPLC) method. Analysis was conducted in triplicate.
$$Entrapment\;efficiency=\left(\frac{Total\;amount\;of\;drug\;(mg)\;-\;free\;drug\;in\;supernatant}{total\;amount\;of\;drug\;(mg)}\right)*100\%$$

#### 4.7.5. Self-Emulsification Time

The self-emulsification time of the BDQ-F-SNEDDS was evaluated to confirm its capability to form a stable formulation. The desired outcome of the SNEDDS formulation is that it readily disperses and forms an emulsion when gently mixed with distilled water. Emulsification time was estimated using USP Apparatus II and 0.5 *w*/*v* SLS solution with constant stirring. Time taken by the formulation to completely disperse in water was estimated [19].

#### 4.7.6. Dilution Studies

BDQ-F-SNEDDS were subjected to dilution in order to check stability without phase separation. BDQ-F-SNEDDS were diluted to 50, 100, and 200-folds with deionized water and gently shaken. The dilutions were further sonicated for 30 s to prevent bubbles, and droplet size, PDI, and % transmittance were recorded [40].

### 4.8. In Vitro Drug Release Study

A drug release study was performed using a dialysis membrane. A pre-activated dialysis membrane with a molecular weight cut-off of 12.4 kDa was loaded with 2 mL of optimized BDQ-F-SNEDDS and BDQ-F suspension, respectively. Two separate 100 mL beakers were filled with 48 mL of 0.1 N HCl medium. The dialysis membrane containing the BDQ-F-SNEDDS and BDQ-F suspension was placed in the respective beakers filled with the medium. The beakers were then placed on a magnetic stirrer at a temperature of 37 °C and a stirring speed of 100 rpm.

Sampling was carried out at specific time intervals (2, 4, 6, 8, 10, and 12 h) by withdrawing 1 mL of sample at each time point. The withdrawn sample volume was replaced with an equal volume of fresh media to maintain sink conditions. HPLC analysis was performed on the collected samples to determine the drug concentration. The cumulative amount of drug release was calculated based on the analysis results. To ensure reliable data, the experiment was conducted in triplicate for statistical significance [41]. 

### 4.9. Stability Studies

The Stability studies were performed according to ICH guidelines [42]. The BDQ-F-loaded SNEDDS carriers were subjected to a physical stability study by storing them for the duration of 6 months under both room temperature and accelerated conditions. At predetermined intervals (0, 3, and 6 months), samples were taken from the storage and analyzed for any observable changes in Physical Appearance, droplet size, PDI, and entrapment efficiency. Additionally, the shelf-life of the optimized BDQ-F-SNEDDS was determined through this analysis [43]. Stability protocol is provided in Appendix A as per industry format.

### 4.10. Cell Toxicity Studies 

The cytotoxicity of both free BDQ-F and optimized BDQ-F-SNEDDS (self-nanoemulsifying drug delivery system) on A549 cells was assessed using the MTT assay. A549 cells were seeded at a density of (5 × 10^3^) cells per well in 96-well tissue culture plates and incubated overnight. Subsequently, the cells were treated with varying concentrations (1–100 µg/mL) of the optimized BDQ-F-SNEDDS, placebo, or BDQ-F. Following the treatment, MTT was added to the wells, and the resulting formazan crystals were solubilized using dimethyl sulfoxide. The absorbance of the solution was measured [44]. The percentage of cytotoxicity, indicating cell viability, was calculated using the provided formula: $$\%\;cytotoxicity\;=\;\frac{Absorbance\;of\;control\;-\;Absorbance\;of\;test}{Absorbance\;of\;control}\;*\;100$$

### 4.11. Statistical Analysis

The experiments were conducted in triplicate, and the data are presented as mean ± standard deviation (SD), with statistical significance set at *p* < 0.05. Data analysis was performed using GraphPad Prism^®^ software, version 6.01.

## 5. Conclusions

Optimized BDQ-F-SNEDDSs were formulated using Design of Experiments (DoE), with Caprylic acid as the lipid and Transcutol P and propylene glycol as the surfactant/cosurfactant system, forming an interfacial film around the oil globules. The prepared SNEDDSs underwent evaluation for various parameters, including droplet size, PDI, and percent transmittance. Key Quality Target Product Profiles (QTPPs) and CQAs were identified to ensure a robust formulation. Risk assessment was performed using an Ishikawa Fish-bone diagram. Optimization of BDQ-F-SNEDDS was accomplished using the Box–Behnken design, resulting in an optimized formulation with a small droplet size and low PdI, indicating stability and robustness. Transmission electron microscopy (TEM) images exhibited a spherical shape and a distinct boundary, suggesting high entrapment efficiency.

Results from in vitro drug release studies have indicated that optimized BDQ-F-SNEDDS displayed greater concentrations of drug release compared to the suspension form. This suggests that the optimized BDQ-F-SNEDDS formulation effectively releases the drug in higher concentrations.

Cell cytotoxicity studies conducted on A549 cells revealed enhanced cellular internalization of BDQ-F-SNEDDS, which may be correlated with improved permeability. The lipophilic oil globules in SNEDDS could potentially enable its penetration through bacterial biofilms, thereby enhancing the effectiveness against resistant strains of bacteria. Furthermore, stability studies indicated that BDQ-F-SNEDDS remained stable at both 40 ± 2 °C and 25 ± 2 °C for up to 6 months. 

The advantages of SNEDDS include its smaller globule size, which facilitates improved drug delivery. Furthermore, it also offers potential benefits for the long-term treatment of multidrug-resistant tuberculosis (MDR-TB) by extending the circulation time of the drug and providing a controlled release pattern. These characteristics hold promise for reducing the dose frequency of administration and enhancing patient compliance.

In summary, BDQ-F-SNEDDS presents a promising strategy for drug delivery that requires further exploration to optimize the treatment protocol for multidrug-resistant tuberculosis (MDR-TB) by decreasing the frequency of dosing and treatment duration. However, it is important to conduct additional research to fully understand the intricate in vivo disposition and pharmacodynamics in animal models.

## Figures and Tables

**Figure 1 antibiotics-12-01510-f001:**
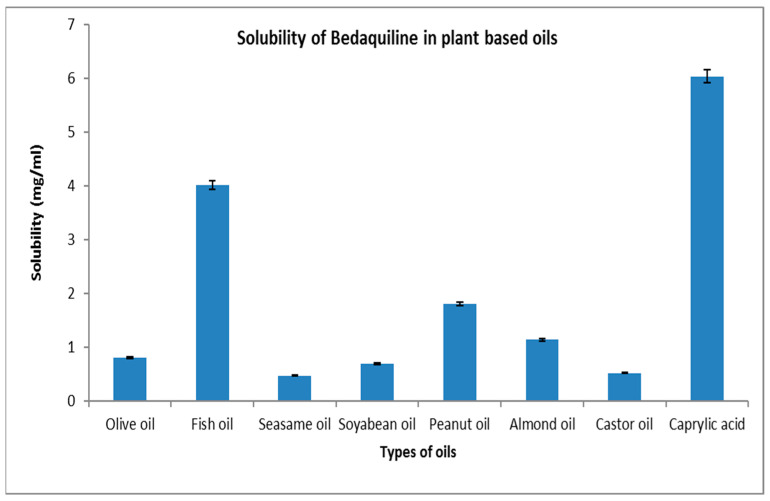
Solubility of BDQ-F in lipids.

**Figure 2 antibiotics-12-01510-f002:**
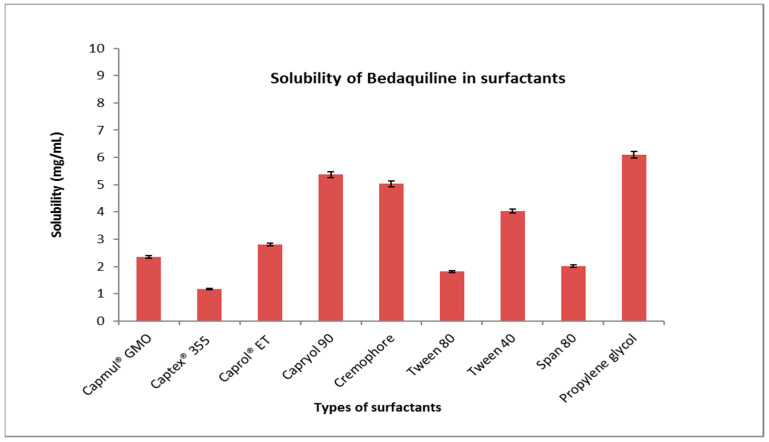
Solubility of BDQ-F in surfactants.

**Figure 3 antibiotics-12-01510-f003:**
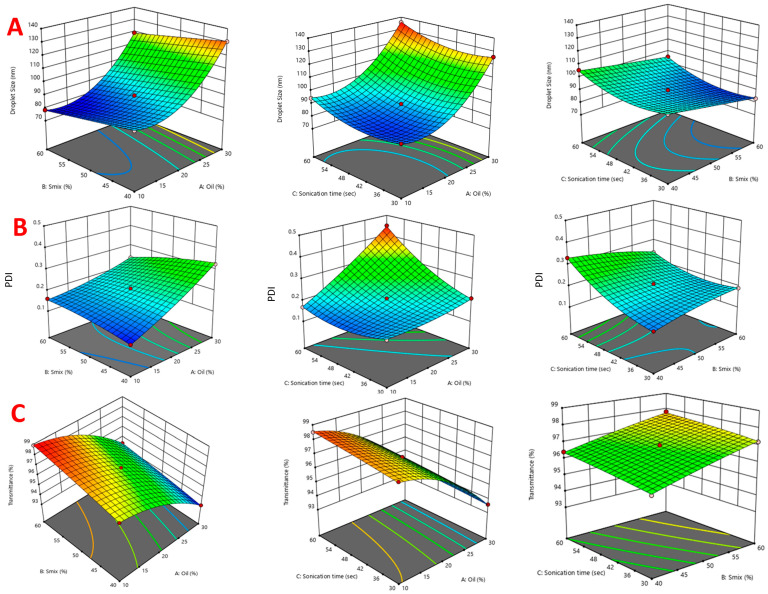
The 3D surface response plot showing effect of independent variables (oil, S_mix_, and sonication time) on dependent variables (**A**) droplet size, (**B**) PDI, and (**C**) Percentage Transmittance.

**Figure 4 antibiotics-12-01510-f004:**
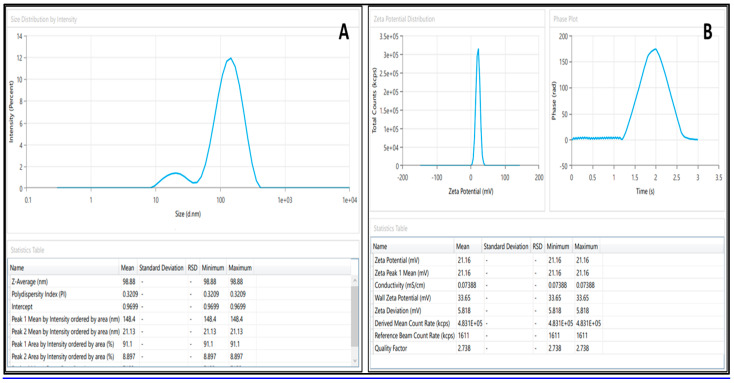
Graphical presentation of (**A**) droplet size, PdI, and (**B**) zeta potential of optimized BDQ-F-SNEDDS.

**Figure 5 antibiotics-12-01510-f005:**
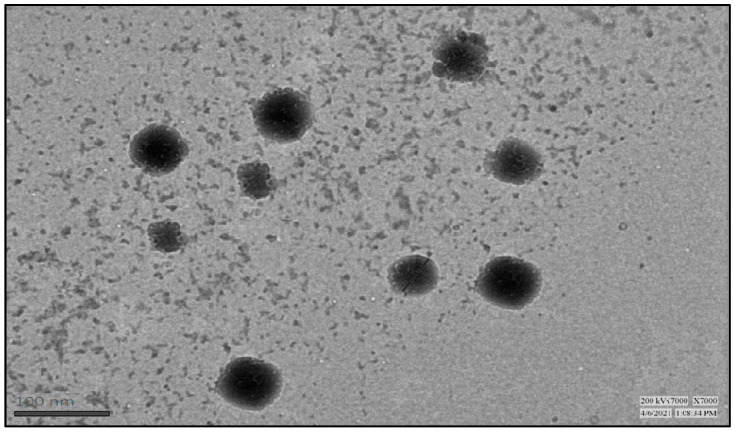
Surface morphology of optimized BDQ-F-SNEDDS droplets by TEM at 100 nm scale.

**Figure 6 antibiotics-12-01510-f006:**
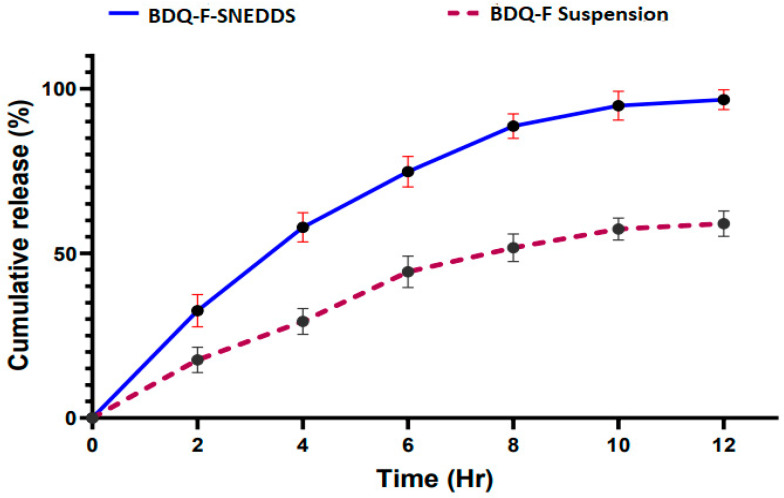
Cumulative (%) drug release from drug suspension and BDQ-F-SNEDDS.

**Figure 7 antibiotics-12-01510-f007:**
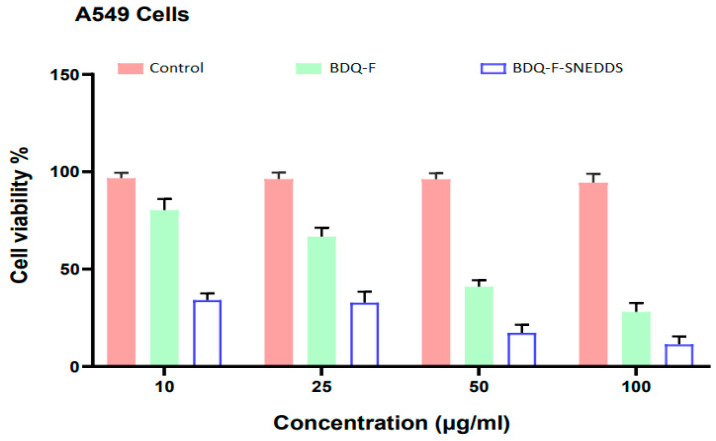
Cytotoxicity studies of BDQ-F-SNEDDS in A549 cell lines at different concentrations; data represented as mean ± S.D., where n = 3.

**Table 1 antibiotics-12-01510-t001:** Effect of dilution on different parameters of BDQ-F-SNEDDS.

Dilution (Folds)	Droplet Size (nm)	PDI	Transmittance (%)
50	99.8	0.34	97.8
100	95.9	0.39	97.1
200	95.1	0.42	96.4

**Abbreviations:** BDQ-F-SNEDDS, bedaquiline-fumarate-self-nano emulsifying drug delivery system; nm, nanometer; PDI, polydispersity index.

**Table 2 antibiotics-12-01510-t002:** Stability results at temperature and humidity of 40 ± 2 °C and 75 ± 5%.

At a Temperature 40 ± 2 °C Humidity 75 ± 5%			
**Testing Parameters**	**Initial**	**3 Months**	**6 Months**
Physical Appearance	Clear	Clear	Clear
Phase Separation	No phase separation	No	No
Caking	No caking	No	No
Size (nm)	98.88 ± 0.19	99.81 ± 0.72	106.04 ± 0.83
PDI	0.34 ± 0.392	0.39 ± 0.321	0.45 ± 0.43
Entrapment Efficiency	98.30%	87.71%	76.20%
**At a Temperature 25 ± 2 °C Humidity 60 ± 5%**			
Physical Appearance	Clear	Clear	Clear
Phase Separation	No phase separation	No	No
Caking	No caking	No	No
Size (nm)	98.88 ± 0.32	99.95 ± 0.45	106.21 ± 0.52
PDI %	0.34 ± 0.392	0.40 ± 0.321	0.45 ± 0.43
Entrapment Efficiency	98.31%	85.60%	75.10%

**Abbreviations:** PDI, polydispersity index.

**Table 3 antibiotics-12-01510-t003:** Variables used for optimization.

Variables	Levels Used	
** *Independent Variables* **	** *Low (−1)* **	** *High (+1)* **
X_1_: Oil (%) X_2_: S_mix_ (%) X_3_: Sonication time (min)	10 30 40 60 30 60	
** *Dependent Variables* **	** *Constraints* **	
Droplet size (nm) PDI Transmittance (%)	Minimize Minimize Maximize	

**Abbreviations:** PDI, polydispersity index.

## Data Availability

The data presented in this study are available on request from the corresponding author. The data are not publicly available due to privacy or ethical restrictions.

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
