# Peer review of "Development of Bedaquiline-Loaded SNEDDS Using Quality by Design (QbD) Approach to Improve Biopharmaceutical Attributes for the Management of Multidrug-Resistant Tuberculosis (MDR-TB)"

_antibiotics, 2023, doi:10.3390/antibiotics12101510_

Round 1
Reviewer 1 Report
In this work, the authors developed a quality by design based self-nanoemulsifying drug delivery system of Bedaquiline, a new treatment for tuberculosis. Below are my comments:
1. Page 2: References are needed for the sentence "..., TB was estimated to be the leading cause of ..."
2. Page 3: References are needed for the sentences "... to optimize isotropic systems has become common practice in both industry and academia." and "... to the traditional one-factor-at-a-time approach."
3. I suggest the authors add a paragrah at the end of the introduction section to be more specific about the goal of this work. This can make the manuscript more easy to follow. Right now, the objectives are hidden in multuple places whereas a more specific paragraph can greatly clarify the goal of this work for readers.
4. Page 5: In the equation of efficiency, it should be "100%" instead of "100". Also be more specific about "amount of drug" (mass or volume?)
5. Page 6, Section 4.1: The authors need to be consistent about the decimal places for these reported data since they represent the accuracy of the experimental equipment can achieve. For example, either it should be "6.3 +/- 10.9" or "6.27+/-10.90". I spotted this issue throughout the manuscript: Section 4.2, Section 4.5.1, Section 4.5.2, Section 4.5.8, Table 3. Please check carefully and correct any of these in the manuscript.
6. Page 8, Section 4.3: Although they are mentioned in the SI, I still suggest the authors explain "A", "B", "C" in these equations. Also, how did the authors derived these equations? If these are from previous studies, please add references. Otherwise, please clarify in the manuscript.
7. Page 9, Figure 3: The resolution is poor. Please replace it with a clear version.
8. Page 10, Section 4.4: It should be "98.17%".
9. Page 10, Section 4.5.3: It should be "Figure 5".
10. Page 13, Section 4.5.9: It should be "Figure 7".
11. Can the authors comment on how to further optimize the efficacy of BDQ besides what have been done in this work?
The English is poor and extensive editing is needed.
Author Response
For reviewer 1
Please see the attachment

Reviewer 2 Report
This study aimed to improve the solubility and therapeutic efficacy of the antibiotic Bedaquiline (BDQ) for drug-resistant TB using a Quality by Design (QbD) approach to develop a self-nano emulsifying drug delivery system. The optimized formulation demonstrated enhanced drug release, stability, and cell killing, indicating its potential as an efficient oral delivery method for TB treatment. Overall it is valuable, however, several concerns need to be addressed.
1. For stability assessment, I would like to see more hash conditions such as heat, freeze-thaw cycles to ensure the stability of the BDQ-F-SNEDDS.
2. A viscosity study is recommended because it is a necessary assessment for SNEDDS regarding its dispersion in the aqueous phase.
3. The methods for measuring drug concentration are very important and should be described in detail in the methods section, such as providing detailed HPLC methods and calibration curve.
Author Response
For reviewer 2
Please see the attachment

Round 2
Reviewer 2 Report
The authors have addressed the issues and I support the publication.